# Social Isolation, Hospitalization, and Deaths from Cardiovascular Diseases during the COVID-19 Epidemic in São Paulo Metropolitan Area in 2020

**DOI:** 10.3390/ijerph191711002

**Published:** 2022-09-02

**Authors:** Lucas Rodrigues de Lima, Paulo Francisco Auricchio da Ponte, Lucca Novais Dias, Marcelo Henrique Lima Silvestre, Paulo Jeng Chian Suen, Antonio de Padua Mansur

**Affiliations:** 1Faculdade de Medicina FMUSP, Universidade de Sao Paulo, Sao Paulo 01246-903, SP, Brazil; 2Serviço de Prevencao, Cardiopatia na Mulher e Reabilitação Cardiovascular, Instituto do Coracao (InCor), Hospital das Clinicas HCFMUSP, Faculdade de Medicina, Universidade de Sao Paulo, Sao Paulo 05403-900, SP, Brazil

**Keywords:** cardiovascular diseases, ischemic heart diseases, cerebrovascular diseases, COVID-19, social isolation, mortality, hospitalizations

## Abstract

Background: The COVID-19 epidemic overloaded the São Paulo metropolitan area (SPMA) health system in 2020. The leading hospitals directed their attention to patients with COVID-19. At the same time, the SPMA Health Secretary decreed social isolation (SI), which compromised the care for cardiovascular diseases (CVD), even though higher cardiovascular events were expected. Methods: This study analyzed mortality from CVD, ischemic heart disease (IHD), and stroke, along with hospital admissions for CVD, IHD, stroke, and SI in the SPMA in 2020. Data regarding hospitalization and mortality from CVD were obtained from the SPMA Health Department, and data regarding SI was obtained from the São Paulo Intelligent Monitoring System. Time-series trends were analyzed by linear regression, as well as comparisons between these trends. Results: there was an inverse correlation between SI and hospitalizations for CVD (R^2^ = 0.70; *p* < 0.001), IHD (R^2^ = 0.70; *p* < 0.001), and stroke (R^2^ = 0.39; *p* < 0.001). The most significant hospitalization reduction was from March to May, when the SI increased from 43.07% to 50.71%. The increase in SI was also associated with a reduction in CVD deaths (R^2^ = 0.49; *p* < 0.001), IHD (R^2^ = 0.50; *p* < 0.001), and stroke (R^2^ = 0.26; *p* < 0.001). Conclusions: Increased social isolation was associated with reduced hospitalizations and deaths from CVD, IHD, and stroke.

## 1. Introduction

On 3 November 2020, the World Health Organization declared COVID-19 a pandemic [1]. With the significant increase in cases and the risk of collapse of the healthcare system, on 21 March 2020, social isolation (SI) was decreed throughout the state of São Paulo [2] to reduce the transmission of the disease. Even with an ongoing pandemic, cardiovascular diseases (CVD) are the leading causes of death in Brazil and the Municipality of São Paulo, representing respectively 27% and 29% of total deaths in 2019, divided mainly into ischemic heart diseases (IHD) and stroke [3]. In this sense, we must consider that hospitalizations and deaths from CVD, IHD, and stroke greatly influence the health system, even in the context of the COVID-19 pandemic, and even more relevantly in the SI period. In 2019, São Paulo city had nearly 200 hospitals, of which 48% were public, and 52% were private [4]. These hospitals, especially those with more than 100 beds, directed their attention preferentially to treating patients with COVID-19. Recent studies have shown a reduction in hospital admissions for CVD during the pandemic in Brazil and other countries [5,6]. Normando et al. showed a 15% reduction in hospitalizations for CVD in Brazil [5] from March to May 2020, and Mafham et al. observed a 40% reduction in hospitalizations for acute coronary syndrome in England in the period from February to March [6]. However, studies were inconclusive regarding deaths from CVD. Brant et al. showed an increase in mortality from CVD in five cities (São Paulo, Fortaleza, Belém, Recife, and Manaus), but a reduction in specific mortality from IHD and stroke in the cities of São Paulo, Fortaleza, and Belém [7]. Therefore, the rates of hospitalizations and deaths from CVD depended on the region analyzed.

In this context, this study analyzed the correlation between SI and hospitalization rates and death from CVD, IHD, and stroke in the Municipality of São Paulo in 2020.

## 2. Materials and Methods

A retrospective study analyzed the trends in hospitalizations and deaths from CVD, IHD, and stroke and their correlation with SI in 2020 during the COVID-19 epidemic in the São Paulo Metropolitan Area (SPMA) from 28 February 2020 to 31 December 2020. We obtained the data on hospital admissions and deaths from CVD, IHD, and stroke from the SPMA from the SPMA Health Department [8]. We grouped CVD under codes I00 to I99, IHD under codes I20 to I25, and stroke under codes I60 to I69. Deaths were considered according to the city of residence. We performed the adjusted mortality rate per 100,000 inhabitants using the direct method based on the 2019 Brazilian population estimate by the Brazilian Institute of Geography and Statistics (IBGE). Daily SI data were obtained from the Intelligent Monitoring System of the city of São Paulo [9].

### Statistical Analysis

We used Excel^®^ [10] to perform the statistical analysis. The level of statistical significance adopted was 5%. To analyze hospitalizations and their correlation with SI in the year 2020, we obtained the monthly values of hospitalizations for CVD, IHD, and stroke, and the percentiles of daily SI were grouped monthly by the arithmetic mean for comparison between SI and hospitalizations. To analyze deaths and their correlation with SI in the year 2020, we grouped weekly, the daily values of deaths from CVD, IHD, and stroke, and the daily SI percentiles were grouped weekly by the arithmetic mean for comparison between events and isolation. Week 1 comprises from 28 February 2020 to 5 March 2020, the date corresponding to the first case of COVID-19 verified in the SPMA, until 31 December 2020, making a total of 44 weeks.

To perform the statistical analysis of hospitalizations, we adopted the monthly SI as an independent variable and hospitalizations for CVD, IHD, and stroke as the dependent variable. We constructed three simple linear regressions, one for each dependent variable. Likewise, we adopted the weekly SI as an independent variable and deaths from CVD, IHD, and stroke as dependent variables. In addition, weekly SI values and CVD, IHD, and stroke deaths throughout the year 2020 (time series) were analyzed by simple linear regression. We compared their slopes using t-statistics and two-tailed t-distribution in Microsoft Excel [11]. For this analysis, we used only data from week four, as the social isolation in the municipality of São Paulo started on 28 February 2020. This selection was defined to prevent bias in the constructed trend lines of the values of social isolation, CVD, IHD, and stroke, measured prior to the decree.

## 3. Results

### 3.1. Social Isolation and Hospitalizations for CVD, IHD, and Stroke

There was an inverse correlation between SI and hospitalizations for CVD (R² = 0.70; y = −1.49x + 108.91; 95%CI= −2.22 to -0.76; *p* < 0.001), IHD (R² =0.70; y = −0.38x + 28.67; IC95%= −0.56 to −0.19; *p* < 0.001) and stroke (R² = 0.39; y = −0.13x + 14.24; IC95% = −0.24 to −0.01; *p* < 0.001) (Figure 1).

The most significant percentage reduction was CVD hospitalization (1.37% decrease in admissions for each increase in SI percentile), followed by IHD (1.31% decrease in hospitalizations for each increase in SI percentile), and stroke (decrease of 0.88% in admissions for each increase SI percentile).

The most significant reduction in the number of hospitalizations for CVD, IHD, and stroke was from March to May 2020, in which social isolation increased from 43.07% to 50.71%. Monthly CVD hospitalizations were reduced by 42.60%, from 52.11 to 29.91 per 100,000 inhabitants; IHD hospitalizations were reduced by 40.46%, from 14.16 to 8.43 per 100,000 inhabitants; and stroke hospitalizations were reduced by 25.10%, from 9.96 to 7.46 per 100,000 inhabitants (Appendix A).

### 3.2. Social Isolation and Deaths from CVD, IHD, and CVD

Data on SI and deaths per 100,000 inhabitants for CVD, IHD, and stroke are shown in the Table 1 below.

There was an inverse correlation between weekly SI and weekly deaths from CVD (R^2^ = 0.49; y = −0.045x + 5.379; 95% CI = −0.059 to −0.031; *p* < 0.001), IHD (R^2^ = 0.50; y = −0.024x + 2.192; IC 95%= −0.031 to −0.016; *p* < 0.001), and stroke (R^2^ = 0.26; y = −0.010x + 1.240; IC 95% = −0.015 to −0.005; *p* < 0.001) (Figure 2).

The most significant percentage reduction was in IHD deaths (1.08% decrease in deaths for each SI percentile increase), followed by CVD (0.84% decrease in deaths for each SI percentile increase), and then by stroke (decrease of 0.80% in deaths for each SI percentile increase).

Throughout 2020, there was a reduction in SI and a progressive increase in weekly deaths during the pandemic period, starting on 03/20/2020 (week 4) (Figure 3A–C).

The weekly SI decreased by 29.1% from March 2020 to December 2020, from 56.4% (week 5) to 40% (week 43) (R^2^ = 0.88; y = −0.35x + 54.251; IC 95% = −0.39 to −0.31; *p* < 0.001). The number of CVD deaths increased by 25.8% from April 2020 to December 2020, from 2.67 in 04/2020 (week 8) to 3.36 deaths per 100,000 inhabitants in 12/2020 (week 41) (R^2^ = 0.22; y = 0.011x + 3.026; IC 95% = 0.004 to 0.018; *p* = 0.002). The number of deaths from IHD increased by 20.2% from March 2020 to December 2020, from 0.99 (week 4) to 1.19 deaths per 100,000 inhabitants (week 41) (R^2^ = 0.22; y = 0.005x + 0.971; IC 95% = 0.002 to 0.009; *p* = 0.002). The number of deaths from stroke increased by 15.3% from March 2020 to December 2020, from 0.72 (week 5) to 0.83 deaths per 100,000 inhabitants (week 43) (R^2^ = 0.10; y = 0.002x + 0.725; IC 95% = 3.4 × 10^−^^5^ to 0.005; *p* = 0.047). There was also a statistically significant difference in the comparisons of linear regression lines between weekly SI and CVD (β1 = 1.67; *p* < 0.001), weekly SI and IHD (β1 = 0.99; 0.001), and weekly SI and stroke (β1 = 0.63; *p* < 0.001).

## 4. Discussion

Regarding the findings of an association of SI with deaths and hospitalizations for CVD, IHD, and stroke, the literature is still scarce. In line with our findings, Wu et al. observed a reduction in the number of hospitalizations in the UK for heart failure by 54% and for myocardial infarction by 32% from the first period of social isolation (23 March 2020) to the nadir for these indices (2 and 4 April 2020, respectively). However, they did not analyze deaths from these same syndromes in this period, as we did in our study for CVD, IHD, and stroke [12].

Another study, carried out by Hu et al., addressed the topic of social health, assessed by the parameters of social isolation, social support, and loneliness, using two cardiovascular risk scores (the atherosclerotic cardiovascular disease risk scale and the Framingham Risk Score) [13]. SI was defined as little contact with family and friends and low engagement in community activities. In this study, a worsening of cardiovascular risk indices were observed, such as the increase in the prevalence of smoking (7.2% of those isolated were active smokers, against 2.2% of those who were non-isolated), arterial hypertension (66.3% in isolated, against 46.5% in non-isolated), lack of physical activity (59.5% showed a sedentary lifestyle among the socially isolated, against 40.1% among the non-isolated), and depressive symptoms (25.3% in isolated, against 9.8% in non-isolated). The other indices showed no statistically significant difference. These unhealthy metrics should be associated with increased CVD mortality, but the authors did not analyze mortality—only CVD risk scores.

A study by Smith et al. looked at the effect of social isolation on fatal cardiovascular events in the UK population and found an increased relative risk of non-hospitalized death from CVD and stroke compared to individuals outside of isolation [14]. However, it should be noted that this study classified social isolation based on three indices: living alone, contact with family and friends, and participation in groups, which differs from our study, in which social isolation was based on the simple permanence of the person in their home, as measured by cell phone location. Therefore, many of these isolated people still had contact with family or friends through social networks and other forms of social interaction.

However, there are still no significant studies that directly quantitatively correlate SI with cardiovascular outcomes in the general population. SI is a complex phenomenon in which factors that increase cardiovascular risk may be present, such as loneliness and low community engagement, a sedentary lifestyle [15,16], and increased hours worked [17,18,19]. In this sense, the reduction in hospitalizations and deaths from cardiovascular diseases may reflect isolation and sanitary conditions, leading fewer people to the health system for fear of contracting COVID-19. On the other hand, social isolation can also promote protective factors, such as reduced work rhythm, less exposure to environmental pollution [20], and an increase adequate sleep [21].

Regarding the lower mortality from CVD, some hypotheses may justify this finding. First, since CVD carriers are a risk group for more severe cases of COVID-19, these patients could have the underlying cause of death recorded as COVID-19 infection, rather than their cardiovascular condition. Another hypothesis would be an improvement in the quality of food. Due to social isolation, people started to eat better, preparing their meals at home and not eating ultra-processed foods as often. Social isolation may also have reduced worker stress related to commuting from home to work.

The main limitations of this study are due to the public records of a population and the data being subject to unpredictable and undetectable errors. In addition, the study was based on a specific population and thus, likely subject to particularities of São Paulo city that are not transferable to other country locations with different population densities, as well as for populations in other countries. Likewise, the more complex healthcare system and different socioeconomic profiles are other conditions which can significantly impact the results observed. Finally, the numbers of daily deaths was obtained based on death certificates, which, however reliable they may be in a metropolis with an organized record system, such as the city of São Paulo, are still subject to errors; for example, some at-home deaths could have been misclassified on the part of the professional who filled in the death certificate.

## 5. Conclusions

Our study showed a significant reduction in hospitalizations and deaths from CVD, IHD, and stroke in São Paulo during the COVID-19 pandemic in 2020. This reduction in hospitalizations and deaths correlated inversely with the rates of SI. Future studies are still needed to determine the causality of this observed phenomenon.

## Figures and Tables

**Figure 1 ijerph-19-11002-f001:**
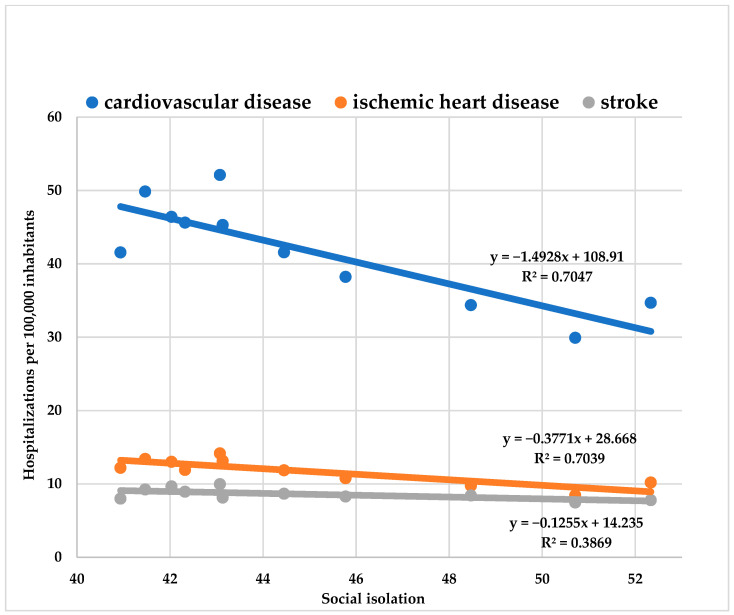
Correlation between weekly social isolation and hospitalization per 100,000 inhabitants from cardiovascular disease, ischemic heart disease, and stroke.

**Figure 2 ijerph-19-11002-f002:**
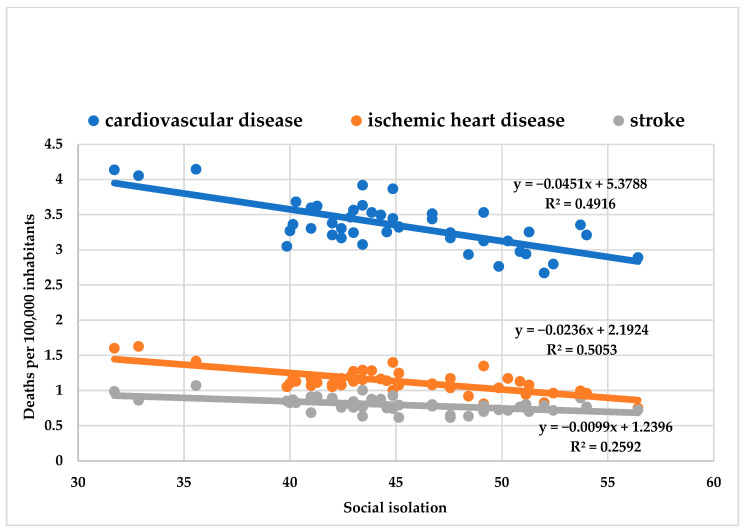
Correlation between weekly social isolation and deaths per 100,000 inhabitants from cardiovascular disease, ischemic heart disease, and stroke.

**Figure 3 ijerph-19-11002-f003:**
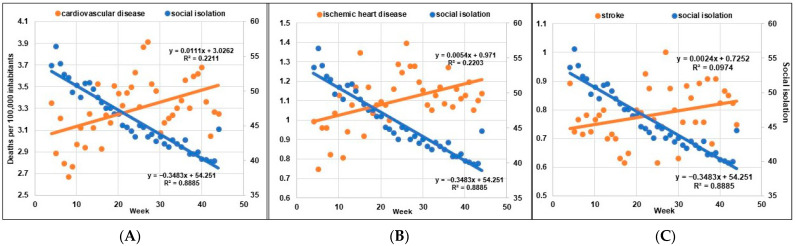
(**A**) Correlation between weekly social isolation and weekly deaths from cardiovascular disease; (**B**) Correlation between weekly social isolation and weekly deaths from ischemic heart disease; (**C**) Correlation between weekly social isolation and weekly deaths stroke.

**Table 1 ijerph-19-11002-t001:** Weekly social isolation (SI) and deaths per 100,000 inhabitants from cardiovascular disease (CVD), ischemic heart disease (IHD), and stroke.

Week	Days		SI (%)	CVD	IHD	Stroke
1	28 February 2020	5 March 2020	32.86	4.05	1.63	0.86
2	6 March 2020	12 March 2020	31.71	4.14	1.60	0.99
3	13 March 2020	19 March 2020	35.57	4.15	1.42	1.07
4	20 March 2020	26 March 2020	53.71	3.35	0.99	0.89
5	27 March 2020	2 April 2020	56.43	2.89	0.75	0.72
6	3 April 2020	9 April 2020	54.00	3.21	0.96	0.77
7	10 April 2020	16 April 2020	52.43	2.80	0.96	0.72
8	17 April 2020	23 April 2020	52.00	2.67	0.83	0.78
9	24 April 2020	30 April 2020	49.86	2.76	1.04	0.72
10	1 May 2020	7 May 2020	50.86	2.97	1.13	0.77
11	8 May 2020	14 May 2020	49.14	3.13	0.81	0.78
12	15 May 2020	21 May 2020	51.14	2.94	0.94	0.80
13	22 May 2020	28 May 2020	51.29	3.25	1.08	0.70
14	29 May 2020	4 June 2020	50.29	3.13	1.17	0.72
15	5 June 2020	11 June 2020	49.14	3.53	1.35	0.70
16	12 June 2020	18 June 2020	48.43	2.93	0.92	0.63
17	19 June 2020	25 June 2020	47.57	3.24	1.17	0.62
18	26 June 2020	2 July 2020	47.57	3.17	1.04	0.65
19	3 July 2020	9 July 2020	46.71	3.51	1.08	0.78
20	10 July 2020	16 July 2020	46.71	3.44	1.10	0.80
21	17 July 2020	23 July 2020	45.14	3.33	1.08	0.79
22	24 July 2020	30 July 2020	44.86	3.45	1.00	0.93
23	31 July 2020	6 August 2020	44.29	3.50	1.16	0.88
24	7 August 2020	13 August 2020	43.43	3.63	1.29	0.77
25	14 August 2020	20 August 2020	45.14	3.32	1.25	0.62
26	21 August 2020	27 August 2020	44.86	3.87	1.40	0.74
27	28 August 2020	3 September 2020	43.43	3.92	1.28	1.00
28	4 September 2020	10 September 2020	43.86	3.53	1.28	0.88
29	11 September 2020	17 September 2020	42.86	3.46	1.20	0.80
30	18 September 2020	24 September 2020	43.43	3.08	1.15	0.63
31	25 September 2020	1 October 2020	42.43	3.17	1.08	0.76
32	2 October 2020	8 October 2020	42.00	3.21	1.05	0.88
33	9 October 2020	15 October 2020	43.00	3.24	1.13	0.84
34	16 October 2020	22 October 2020	42.43	3.30	1.17	0.76
35	23 October 2020	29 October 2020	42.00	3.38	1.09	0.89
36	30 October 2020	5 November 2020	43.00	3.56	1.27	0.76
37	6 November 2020	12 November 2020	41.00	3.30	1.07	0.91
38	13 November 2020	19 November 2020	41.00	3.60	1.16	0.68
39	20 November 2020	26 November 2020	41.29	3.62	1.11	0.91
40	27 November 2020	3 December 2020	40.29	3.68	1.13	0.83
41	4 December 2020	10 December 2020	40.14	3.36	1.20	0.87
42	11 December 2020	17 December 2020	39.86	3.05	1.05	0.85
43	18 December 2020	24 December 2020	40.00	3.27	1.10	0.83
44	25 December 2020	31 December 2020	44.57	3.25	1.14	0.75

## Data Availability

The datasets used and/or analyzed during the current study are available from the corresponding author on reasonable request.

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
