# Peer review of "Social Isolation, Hospitalization, and Deaths from Cardiovascular Diseases during the COVID-19 Epidemic in São Paulo Metropolitan Area in 2020"

_ijerph, 2022, doi:10.3390/ijerph191711002_

Round 1
Reviewer 1 Report (Previous Reviewer 1)
The paper can be accepted. Thank you to any hours for their response
Author Response
Our study showed a significant reduction in hospitalizations and deaths from CVD, IHD, and stroke in São Paulo during the COVID-19 pandemic in 2020. This reduction in hospitalizations and deaths correlated inversely with the rates of SI. Future studies are still needed to determine the causality of this observed phenomenon.
This manuscript is a resubmission of an earlier submission. The following is a list of the peer review reports and author responses from that submission.
Round 1
Reviewer 1 Report
The paper entitled "Social isolation, hospitalization, and deaths from 2 cardiovascular diseases in the COVID-19 epidemic in São 3 Paulo Metropolitan Area in 2020" is an interesting uptake on the effect of social isolation and cardiovascular disease hospitalization.
The paper points out quite a few interesting angles specially factors that are defining social issues such as isolation.
However, there are a few revisions that need to be addressed to understand the paper.
Major revisions
a)It will be interesting to see if the decrease in hospitalization during social isolation has a correlation with socioeconomic status. Is there a reduction in hospitalization in a certain section of the society or is it a general evidence?
b) Does sex play a role in the hospitalization number ?
c) is there any data available about hospitalization, and the post-vaccination phase? did the number jump back or do we still see a reduced number of hospitalization?
Minor revisions
a) There are some minor spell check
Author Response
Thanks to the reviewer for the time spent reviewing my manuscript.
Reviewer 1
Comments and Suggestions for Authors
The paper entitled "Social isolation, hospitalization, and deaths from 2 cardiovascular diseases in the COVID-19 epidemic in São 3 Paulo Metropolitan Area in 2020" is an interesting uptake on the effect of social isolation and cardiovascular disease hospitalization.
The paper points out quite a few interesting angles specially factors that are defining social issues such as isolation.
However, there are a few revisions that need to be addressed to understand the paper.
Major revisions
- a) It will be interesting to see if the decrease in hospitalization during social isolation has a correlation with socioeconomic status. Is there a reduction in hospitalization in a certain section of the society or is it a general evidence?
RE: It is general evidence. We do not have data on socioeconomic status, as it is a study of the population as a whole. I believe that variations could occur according to socioeconomic level, but as we do not have this data, I included it as a limitation of the study.
- b) Does sex play a role in the hospitalization number?
RE: We do not have data on sex hospitalization numbers.
- c) is there any data available about hospitalization, and the post-vaccination phase? did the number jump back or do we still see a reduced number of hospitalization?
RE: We do not have those data because vaccination in Brazil started in January 2021. On January 17th, 2021, vaccination occurred in only 112 subjects. We do not analyze the post-vaccination phase and the number of hospitalizations.
Minor revisions
- There are some minor spell check
RE: OK. We reviewed the English.
Reviewer 2 Report
Line 16 abbreviation CV
Line 17 and 18 is the same abbreviation for cardiovascular disease and cerebrovascular
Line 23 p value for stroke has semicolon extra
Line 28 two times CVD abbreviation is appearing
Line 29 add Sao Paolo and Brazil into keywords
Throughout the text, spacing between the last word and bracket for reference is missing. For example lines 33,35,39,44,61 etc.
Remove punctuation mark before brackets with references. For example Lines 43, 46, 50,61 etc.
Line 58 explain abbreviation DIC
Line 58 and SI should be “and their correlation with SI”
Lines 59, 75 and 87 Date writings are inconsistent. Decide on 1 format.
Line 59. There is no punctuation mark behind the date unless it’s the end of the sentence but then new sentence is supposed to start with the capital letter
Line 88 is CDbV new abbreviation? If yes explain it
Line 94 extra semicolon with p value
Figure 1 is missing label and description
Why haven’t the authors provided accompanying data for Figure 1 as supplementary material
Lines 101-106 Instead of description make a graph that shows on the left side of y-axis hospitalization, right side of y-axis SI and x-axis months
Line 110 Table number is missing
Table is missing label and description
Figure 2 is missing label and description
Figure 3 is hard to read. It needs better resolution or break it into 3 separate figures.
Lines 167 and 168 Two or more references write into same bracket using comma. For example: [14,15]
"In the Introduction section, the authors stated that CVDs are the main cause of death in Brazil. But since the Brazil is a huge country, it would be more informative for readers if additional data on CVDs in Sao Paolo region are also given. In addition, please provide information on the total number of hospitals in Sao Paolo, are the hospitals included in this study only ones to treat people with CVD problems, were these public or private hospitals? Can you inform the readers what were average hospitalization and deaths caused by CVD-related health problems prior to COVID-19 situation, let’s say in 2018? Were these pre-COVID trends linear or they changed throughout the year"
Author Response
Thanks to the reviewer for the time spent reviewing my manuscript.
Reviewer 2
Comments and Suggestions for Authors
Line 16 abbreviation CV
RE: corrected
Line 17 and 18 is the same abbreviation for cardiovascular disease and cerebrovascular
RE: corrected
Line 23 p value for stroke has semicolon extra
RE: corrected
Line 28 two times CVD abbreviation is appearing
RE: corrected
Line 29 add Sao Paolo and Brazil into keywords
RE: Keywords OK
Throughout the text, spacing between the last word and bracket for reference is missing. For example lines 33,35,39,44,61 etc.
RE: corrected
Remove punctuation mark before brackets with references. For example Lines 43, 46, 50,61 etc.
RE: done
Line 58 explain abbreviation DIC
RE: corrected. DIC is the abbreviation of IHD in portuguese
Line 58 and SI should be “and their correlation with SI”
RE: done
Lines 59, 75 and 87 Date writings are inconsistent. Decide on 1 format.
RE: done
Line 59. There is no punctuation mark behind the date unless it’s the end of the sentence but then new sentence is supposed to start with the capital letter
RE: done
Line 88 is CDbV new abbreviation? If yes explain it
RE: corrected. CDbV (or AVC) is the abbreviation of stroke in Portuguese.
Line 94 extra semicolon with p value
RE: corrected
Figure 1 is missing label and description
RE: done
Why haven’t the authors provided accompanying data for Figure 1 as supplementary material
RE: Done
Lines 101-106 Instead of description make a graph that shows on the left side of y-axis hospitalization, right side of y-axis SI and x-axis months
RE: We think the description is more informative.
Line 110 Table number is missing
RE: It is only one table. No number is needed.
Table is missing label and description
RE: Done
Figure 2 is missing label and description
RE: Done
Figure 3 is hard to read. It needs better resolution or break it into 3 separate figures.
RE: OK. We improved the quality of Figure 3.
Lines 167 and 168 Two or more references write into same bracket using comma. For example: [14,15]
RE: OK
"In the Introduction section, the authors stated that CVDs are the main cause of death in Brazil. But since the Brazil is a huge country, it would be more informative for readers if additional data on CVDs in Sao Paolo region are also given. In addition, please provide information on the total number of hospitals in Sao Paolo, are the hospitals included in this study only ones to treat people with CVD problems, were these public or private hospitals? Can you inform the readers what were average hospitalization and deaths caused by CVD-related health problems prior to COVID-19 situation, let’s say in 2018? Were these pre-COVID trends linear or they changed throughout the year"
RE: Additional data on CVDs in the São Paulo region and hospital details were included in the Introduction section.
Reviewer 3 Report
This study analyzed mortality from CVD, ischemic heart disease (IHD), cerebrovascular
diseases (CVD), hospital admissions for CVD, IHD, stroke, and SI in Sao Paulo metropolitan area in the year 2020. The authors showed that there was an inverse correlation between IS and hospitalizations for CVD (R2=0.70; p<0.001), IHD ( R2=0.70; p<0.001), and stroke (R2=0.39; p<;0.001). The greatest reduction in hospitalizations was from March to May, a period in which the SI increased from 43.07% to 50.71%. The increase in SI was also associated with a reduction in CVD deaths (R2=0.49; p<0.001), IHD (R2=0.50; p<0.001),
and stroke (R2=0.26; p<0.001). The findings of the study are interesting. I have a few comments for the authors:
-
The authors could add to the limitations that there is potential for results to be biased as the deaths at home could have been misclassified.
-
The results of the study may not be applicable to other populations. In fact, as authors mention, increased mortality was noted in other populations.
Author Response
Thanks to the reviewer for the time spent reviewing my manuscript.
Reviewer 3
Comments and Suggestions for Authors
This study analyzed mortality from CVD, ischemic heart disease (IHD), cerebrovascular diseases (CVD), hospital admissions for CVD, IHD, stroke, and SI in Sao Paulo metropolitan area in the year 2020. The authors showed that there was an inverse correlation between IS and hospitalizations for CVD (R2=0.70; p<0.001), IHD ( R2=0.70; p<0.001), and stroke (R2=0.39; p<;0.001). The greatest reduction in hospitalizations was from March to May, a period in which the SI increased from 43.07% to 50.71%. The increase in SI was also associated with a reduction in CVD deaths (R2=0.49; p<0.001), IHD (R2=0.50; p<0.001), and stroke (R2=0.26; p<0.001). The findings of the study are interesting. I have a few comments for the authors:
- The authors could add to the limitations that there is potential for results to be biased as the deaths at home could have been misclassified.
RE: OK. Done
- The results of the study may not be applicable to other populations. In fact, as authors mention, increased mortality was noted in other populations.
RE: Yes, we agreed and we stated in the limitations of the study section.